# Pulmonary Function and Associated Prognostic Factors in Children After COVID-19: A Retrospective Cohort Study

**DOI:** 10.3390/medicina61122136

**Published:** 2025-11-29

**Authors:** Mega Septiana, Nastiti Kaswandani, Irene Yuniar, Adhi Teguh Perma Iskandar, Henny Adriani Puspitasari, Hindra Irawan Satari

**Affiliations:** Department of Child Health, Faculty of Medicine, Universitas Indonesia—Cipto Mangunkusumo Hospital, Jakarta 10430, Indonesia; mega.septiana01@ui.ac.id (M.S.); ireneyuniar1@gmail.com (I.Y.); adhitpi@gmail.com (A.T.P.I.); henny.adriani01@ui.ac.id (H.A.P.); hsatari@ikafkui.net (H.I.S.)

**Keywords:** pediatric, children, COVID-19, respiratory function, long COVID, spirometry

## Abstract

*Background and Objectives*: Reports of respiratory function in COVID-19 survivors are still rare, especially in children. This study aims to determine the prevalence and prognostic factors that influence long-term respiratory function in children after COVID-19. *Materials and Methods*: An observational analytical study with a retrospective cohort design was conducted between January and June 2024. The subjects were pediatric patients aged 5–18 years with confirmed history of COVID-19. Respiratory function was evaluated with spirometry. The analyzed prognostic factors included clinical classification of COVID-19, gender, age, comorbidities, history of ventilator support, history of hospitalization and persistent symptoms. *Results*: A total of 100 subjects were included in this study. The subjects were 53% female, 52% aged ≥ 12–18 years, and 76% had at least one comorbidity, the most common being obesity (27%). The majority (73%) had a history of mild COVID-19, and 78% were not hospitalized. The prevalence of impaired lung function was 47%, dominated by restrictive lung pattern. The prevalence of long COVID was 18%, with the most common symptom being fatigue (13%). The presence of persistent symptom is significantly associated with abnormal spirometry result (*p* = 0.03, RR 1.99; 95% CI 1.38–2.87). Undernourished status and moderate-to-severe and critical COVID-19 significantly influence long-term respiratory function with *p* = 0.002, aOR 5.64; CI 95% 1.89–16.85 and *p* = 0.006, aOR 5.18; and CI 95% 1.59–16.89, respectively. *Conclusions*: The prevalence of impaired lung function in children after COVID-19 was 47%. Persistent symptoms, undernourished status, and moderate-to-critical severity of COVID-19 were found to be associated with impaired long-term respiratory function in post-COVID-19 pediatric patients. Further prospective studies are needed to confirm these findings and clarify causal mechanisms.

## 1. Introduction

Coronavirus disease 2019 (COVID-19) is an acute respiratory infectious disease caused by severe acute respiratory syndrome coronavirus 2 (SARS-CoV-2) that first appeared in Wuhan, China, at the end of 2019. In March 2020, the World Health Organization (WHO) declared COVID-19 to be a global pandemic [1].

In children, COVID-19 occurred in a smaller proportion than adult (<2%), and most cases were mild [2,3,4]. In Indonesia, the proportion of children with COVID-19 was 13.8%. Mortality rate of COVID-19 in Indonesian children was also higher (1.2%) compared to the global mortality rate (0.48%) [3,4,5,6].

As the cumulated incidence of COVID-19 increases, a growing concern arises on the impact on the respiratory system after acute infection [4]. Persistent impairment of pulmonary function and exercise capacity have been known to last for months or years in survivors from other coronavirus pneumonia (SARS and MERS), but still little information is available regarding SARS-CoV-2 [7,8,9]. Although adult studies of post-COVID-19 sequelae demonstrate restrictive lung disorders and impaired diffusion capacity as common long-term outcomes [10,11,12], reports in children remain limited and sometimes contradictory. Some pediatric studies found normal lung function after SARS-CoV-2 infection [9,13,14], while others reported restrictive or obstructive patterns and diffusion impairment, particularly in children with more severe symptoms [8,10]. This inconsistency highlights the need for further pediatric-focused research.

Spirometry is the most widely used tool to evaluate pulmonary function in both adults and children. It can differentiate between restrictive and obstructive ventilatory disorders, providing important insights into lung sequelae following respiratory infections [15]. Restrictive lung disease, which is commonly reported post-COVID-19, is characterized by reduced lung expansion and forced vital capacity (FVC), while obstructive disorders involve increased airway resistance [15,16]. Such functional changes may be linked to alveolar damage, chronic inflammation, and possible fibrotic remodeling after viral pneumonia [10,12,17].

Restrictive-type lung disorder is characterized by reduced distensibility of the lungs, compromising lung expansion, and, in turn, reduced lung volumes, particularly with reduced total lung capacity (TLC). Restrictive pattern in spirometry was defined by decreased FVC (≤80% predicted value), normal or decreased FEV1 (≤80% predicted value), and normal or increased ratio FEV1 [15]. Meanwhile, obstructive-type lung disorder is characterized by obstruction of respiratory flow and is often accompanied with increased resistance of airway [16].

Patophysiologically, COVID-19 causes the most damage to the lungs, including alveoli epithel, hyaline membrane, and capillary and alveoli septum [10]. This indicates potential chronic remodeling of the blood vessels and alveolus that eventually will lead to lung fibrosis and/or pulmonary hypertension [12,17]. Lung fibrosis will impair lung function and is considered as a restrictive pattern in spirometry. In addition, the functional loss of respiratory muscles caused by excessive use due to cough and dyspnea will also cause restrictive-type lung disorder [10].

Nutritional status is another important factor influencing pulmonary outcomes. Undernutrition during growth can impair lung development, increase susceptibility to respiratory infections, and predispose children to long-term restrictive lung disorders [18,19,20]. Cohort studies have shown that children with a history of severe malnutrition demonstrate reduced lung capacity later in life [18], while nutritional deficits in early childhood are associated with impaired lung parenchyma development [19]. Conversely, obesity has also been discussed as a comorbidity in COVID-19, although its impact on lung function post-infection appears less consistent [20].

In addition to nutritional status, persistent symptoms—often referred to as “long COVID” or “post-acute sequelae of COVID-19 (PASC)”—have been increasingly recognized in children. The symptoms that persist for 12 weeks can be labeled as long COVID-19 or post-acute sequelae of COVID-19 (PASC). A plethora of unspecific symptoms that present after 4 weeks after confirmed SARS-CoV-2 infection without any other medical explanation [21]. Several mechanisms that might explain long COVID-19 include persistence of virus and/or viral components, virus-induced tissue damage, endothelial dysfunction, coagulopathy, autonomic dysfunction, chronic inflammation, and autoimmunity [21].

Studies report prevalence ranging from 14 to 25%, with fatigue, respiratory complaints, and neurocognitive issues being the most common symptoms [21,22,23,24]. These symptoms may be linked to underlying physiological impairments, including reduced lung function, although the relationship remains debated [25,26].

Given the rising incidence of pediatric COVID-19 and the scarcity of long-term outcome data in children, especially from low- and middle-income countries, this study aimed to describe lung function in children after COVID-19 and to determine the prevalence, characteristics, and prognostic factors influencing long-term respiratory function. We hypothesize that children may experience impaired pulmonary function following COVID-19 infection, with undernutrition and a more severe disease course acting as major risk factors for long-term respiratory sequelae.

## 2. Materials and Methods

### 2.1. Study Design

We conducted an observational analytical study with a retrospective cohort design between January and May 2024 at Cipto Mangunkusumo Hospital, Jakarta, Indonesia. Children with a confirmed history of COVID-19 were identified from hospital records and invited for follow-up evaluation. Eligible patients with confirmed COVID-19 between January 2020 and December 2023 were identified from hospital records, and follow-up spirometry evaluation was performed in 2024 after ethical approval (issued in late 2023).

### 2.2. Participants

The eligible participants were pediatric patients aged 5–18 years who had a confirmed diagnosis of COVID-19 (RT-PCR or antigen positive) between January 2020 and December 2023 and had recovered at least 6 months prior to the study. Written informed consent was obtained from parents or legal guardians.

The exclusion criteria were as follows:Pre-existing chronic lung disease (e.g., bronchopulmonary dysplasia, lung hypoplasia, lung cysts, or congenital heart disease).Documented decreased lung function prior to COVID-19 infection, assessed by medical records or parental reports of persistent respiratory symptoms or abnormal spirometry before SARS-CoV-2 infection (when available).Underlying disease known to affect long-term lung function (e.g., severe persistent asthma diagnosed according to GINA criteria, confirmed by medical records from pediatric pulmonologists or relevant specialists).

Nutritional status was determined using WHO BMI-for-age z-scores: <−2 SD classified as undernourished, >+2 SD classified as obese, and −2 SD to +2 SD classified as normal.

### 2.3. Respiratory Evaluation

#### 2.3.1. Clinical Classification of COVID-19

The severity of the initial COVID-19 episode was classified based on medical records according to the WHO and the Indonesian Pediatric Society criteria:Asymptomatic: Positive test without clinical symptoms.Mild: Presence of mild respiratory symptoms without hypoxemia.Moderate: Pneumonia without hypoxemia or with mild hypoxemia not requiring oxygen therapy.Severe: Pneumonia with severe respiratory distress or hypoxemia requiring oxygen therapy.Critical: Requiring intensive care, mechanical ventilation, or organ support.

#### 2.3.2. Spirometry

Pulmonary function was assessed using a spirometer (HI-901, Chest MI, Inc., Tokyo, Japan) in accordance with the American Thoracic Society/European Respiratory Society (ATS/ERS) 2019 recommendations. Each subject performed at least three technically acceptable maneuvers, and the highest values were recorded. All pulmonary function parameters were evaluated: forced vital capacity (FVC), forced expiratory volume in one second (FEV_1_), forced expiratory flow at 25% of FVC (FEF_25_), and forced expiratory flow at 50% of FVC (FEF_50_). The predicted values were adjusted for age, sex, and height [27].

Spirometry results were categorized as follows:Normal: FEV1 and FVC ≥ 80% predicted.Obstructive: FEV1 < 80% predicted with FEV1/FVC ratio reduced.Restrictive: FVC < 80% predicted with preserved or elevated FEV1/FVC ratio.Mixed: both FEV1 and FVC < 80% predicted with reduced FEV1/FVC ratio.

The participants were scheduled for spirometry at least 6 months after recovery. Eligible children were contacted through outpatient follow-up clinics and invited for evaluation.

### 2.4. Statistical Analysis

The data were analyzed using IBM SPSS Statistics version 25 (IBM Corp., Armonk, NY, USA). The descriptive statistics were presented as frequencies and percentages for categorical variables and as means or medians for continuous variables. For categorical variables, chi-square test was used; if the expected count was <5, Fisher’s exact test was applied. A *p*-value < 0.05 was considered statistically significant. Variables with *p* < 0.25 in bivariate analysis were included in multivariate logistic regression. Adjusted odds ratios (aORs) with 95% confidence intervals (CIs) and relative risks (RRs) were reported for significant predictors.

### 2.5. Ethical Approval

This study was approved by the Ethics Committee of the Faculty of Medicine, Universitas Indonesia—Cipto Mangunkusumo Hospital (No. KET-1850/UN2.F1/ETIK/PPM.00.02/2023). Written informed consent was obtained from all participants and/or their legal guardians.

## 3. Results

A total of 121 patients met the inclusion criteria. However, 21 patients were excluded due to underlying congenital heart disease (*n* = 13) or chronic lung disease (*n* = 8). Therefore, a total of 100 subjects were included in the final analysis. No missing data or loss to follow-up occurred (Figure 1).

Of the 100 subjects, 53% subjects were female and 52% were aged ≥ 12–18 years. The majority (76%) had comorbidity, and 20 subjects had at least two comorbidities. The most common were obesity (27%), undernourishment (19%), and chronic kidney disease (16%). Of the 19 subjects who were undernourished, only one was categorized as having severe malnutrition. In total, 73 subjects had a history of mild COVID-19 and only 4 subjects had critical COVID-19. None had recurrent COVID-19. The majority (78%) was not hospitalized, and 65 subjects had COVID-19 12–36 month before undergoing spirometry procedure. The demographic and clinical characteristics of the participants are shown in Table 1.

### 3.1. Pulmonary Function

Several parameters were used to assess lung function, including FEV1/FVC, FEV1 (%), FVC (%), FEF50 (%), and FEF25 (%). The mean percentage of predicted values for all parameters was more than 80%, except for FEV1, which had a median of 77.9%. Additional data are provided in Appendix A.

Spirometry results were normal in 53 subjects, while 47 subjects showed abnormal result—46 with a restrictive pattern and 1 with an obstructive pattern. The subject with an obstructive pattern had mild persistent asthma but no spirometry results prior to the study.

Detailed spirometry parameters are presented in Appendix A. In brief, the mean predicted values of most parameters were above 80%, except for FEV1, which showed a median of 77.9%, indicating a mild reduction in expiratory flow in a proportion of subjects.

### 3.2. Persistent Symptoms

Out of 100 subjects, 18% reported persistent symptoms after recovering from COVID-19, the most common being fatigue, as shown in Figure 2. There were 12 subjects that reported more than one persistent symptom (details provided in Appendix A). The symptoms were reported to persist for more than 12 weeks. The presence of persistent symptoms after COVID-19 was significantly associated with abnormal spirometry results (*p* = 0.03, RR 1.99, 95%, and CI 1.38–2.87), whereas the number of persistent symptoms was not associated with abnormal spirometry results. However, out of 12 subjects with more than one persistent symptom, there were 11 who had abnormal spirometry results (Appendix A).

### 3.3. Prognostic Factors for Long-Term Respiratory Function

A statistically significant result was found between abnormal spirometry results and the clinical classification of COVID-19 (*p* = 0.002, RR 1.96; 95% CI 1.35–2.86) and undernourished state (*p* = 0.007, RR 1.86; 95% CI 1.27–2.73) (Table 2). Among the study population, 19% were undernourished and 23% had experienced moderate-to-critical COVID-19 symptoms. Both factors were significantly associated with impaired spirometry results in the multivariate analysis. From the multivariate analysis, we found that moderate-to-critical COVID-19 severity and undernourishment were significantly associated with impaired spirometry results in children after COVID-19 (Table 3). Clinical classification referred to the initial severity of COVID-19 (asymptomatic/mild vs. moderate-to-critical). A significant association was observed (*p* = 0.006), with children who experienced moderate-to-critical COVID-19 showing more than fivefold higher odds of impaired pulmonary function compared with those who had asymptomatic or mild disease.

## 4. Discussion

In our study, 76% of the subjects had comorbidity, 20 of whom had ≥2 comorbidities. This result was higher than that reported in a prospective study in Australia, in which 18.9% of children with confirmed COVID-19 had comorbidities [7]. However, in that study, obesity and undernourishment were not considered comorbidities. Meanwhile, in our study, the most common comorbidities are obesity (27%), undernourishment (19%), and chronic kidney disease (16%). The high number of subjects with comorbidities was probably because our study was conducted at the national referral hospital of Indonesia.

Our study showed that most subjects had a history of mild COVID-19 (73%): 4% were asymptomatic, 16% had moderate disease, 3% had severe disease, and 4% were in a critical state requiring ventilator support. This finding is consistent with global reports indicating that most children had mild COVID-19. Over three-quarters of the subjects (78%) were not hospitalized when diagnosed with COVID-19. As most of them had a mild COVID-19, hospitalization was not required, and they only needed to self-quarantine.

### 4.1. Pulmonary Function

Our study found that 53% of the subjects had normal lung function, 46% had a restrictive pattern, and 1% had an obstructive pattern. The subject with an obstructive pattern had been diagnosed with mild persistent asthma prior to the study but had never undergone spirometry. Therefore, it is difficult to determine their relation with COVID-19.

Our findings were similar to those of Onay et al. in Türkiye, who reported that 60% of children with a history of confirmed COVID-19 had normal spirometry results [8]. They reported a much lower proportion with a restrictive pattern (27.2%), while 10.9% of subjects showed an obstructive pattern and 1.8% showed a mixed pattern. Another study, by Bottini et al., which performed spirometry on children with a history of mild-to-moderate COVID-19 1 month after infection, found that all the participant had normal spirometry results [9]. Ozturk et al. conducted a conducted a study three months after SARS-CoV-2 infection and reported significant diffusion impairment in children with a history of severe COVID-19 [10]. Unfortunately, diffusion-related parameters such as blood gas analysis and diffusing capacity of the lungs for carbon monoxide (DLCO) were not performed in our study. The predominance of a restrictive pattern was also observed in studies involving adult participants. Salem et al. reported that more than half of adult COVID-19 survivors had restrictive lung function, and more than one-third had diffusion impairment [11]. Similarly, a meta-analysis by Castro et al. in adult populations reported a predominance of restrictive lung patterns (15%), followed by obstructive lung patterns (7%) [12].

Despite these studies, Dobkin et al. reported different findings. Their prospective cohort study, which included 29 children with a mean age of 13.1 ± 3.9 years, found that only 3 had an obstructive pattern and none had a restrictive pattern on spirometry [13]. A meta-analysis by Bakhtiari et al. also found no evidence of pulmonary function impairment in children following SARS-CoV-2 infection. They further reported that the clinical classification of COVID-19 and asthma comorbidity were not associated with pulmonary function in children after infection [14].

When compared with the general pediatric population without a history of COVID-19, the prevalence of abnormal spirometry in our cohort (47%) was considerably higher. Previous community-based studies in healthy school-aged children have reported abnormal lung function in fewer than 10% of cases, mostly related to undiagnosed asthma or transient respiratory infections rather than restrictive patterns [21,22,23,24]. This contrast strengthens the notion that SARS-CoV-2 infection may contribute to an increased risk of restrictive lung impairment the pediatric population.

### 4.2. Persistent Symptoms

The proportion of subjects with persistent symptoms in our study was 18%, and 17 out of 18 of these subjects reported more than one symptom. This finding consistent with a cohort study in Italy, which reported that 14.7% of children experienced persistent symptoms for several months after being infected with SARS-CoV-2 [22]. In contrast, a meta-analysis by Leon et al. reported a higher prevalence of long COVID in children (25.4%), with the most common persistent symptoms being mood disorders (16.5%), fatigue (9.6%), and sleep disturbances (8.4%) [23].

Another study by Knoke et al. in Germany found that 36% children reported persistent symptom after symptomatic COVID-19, with the most frequent are fatigue (20%), breathing disturbance (16%), anosmia/ageusia (16%), cough (8%) and headache (4%) [24]. Children with asymptomatic COVID-19 also reported persistent symptoms, although at a lower rate (22%), with the most common being fatigue (11.1%), breathing difficulties (4.4%), headache (4.4%), and anosmia/ageusia (6.6) [24]. In contrast, in our study, none of the subjects with asymptomatic COVID-19 reported persistent symptom.

Similarly to previous studies, our study found that fatigue was the most frequent symptom persisting after COVID-19. Fatigue was described as reduced tolerance for physical activity or other daily tasks typically performed by the subjects. This symptom may be caused to decreased FVC observed in restrictive-type lung disorders [25,27].

We also found that persistent symptom was associated significantly with abnormal spirometry results (*p* = 0.03). This finding is consistent with that of Manglani et al., who reported that persistent dyspnea after COVID-19 in adult subjects was associated with restrictive-type lung disorders [25]. However, Bode et al. found no significant correlation between persistent symptom and spirometry results in adults and children with a history of mild COVID-19 [26]. These differing results may be due to variations in participants’ characteristics, such as the clinical classification of COVID-19 and the presence of comorbidities. In our study, 11 out of 12 subjects with more than one persistent symptom had abnormal spirometry results. Therefore, the statistically insignificant result may be due to the small sample size for this variable and potential limitations in data collection. Almost all subjects did not report their symptoms during routine hospital visits, resulting in a lack of detailed data regarding the number and type of persistent symptoms after COVID-19.

### 4.3. Prognostic Factors for Long-Term Respiratory Function

From multivariate analysis, we found that moderate-severe-critical COVID-19, as well as undernourished status, were significantly associated with impaired spirometry results in children after COVID-19 (*p* = 0.006, aOR 5.18; 95% CI 1.59–16.89 and *p* = 0.002, aOR 5.4; and 95% CI 1.89–16.85, respectively). Our findings underscore the importance of early identification of at-risk children. Future longitudinal studies with baseline and follow-up pulmonary function assessments are encouraged to confirm these associations and explore underlying mechanisms.

In this study, the clinical classification of moderate-to-severe and critical COVID-19 showed a significant influence on long-term respiratory function in children. This aligns with findings by Sudre et al., who reported that patients experiencing more than five symptoms during the first week of illness or with a hospitalization history were more likely to develop long COVID [28]. Another study also found that intensive care admission—an indicator of severe or critical COVID-19—was associated with post-infection lung function impairment [29].

In contrast, a single-center study in India by Sharanya et al. reported different findings. They found that the clinical classification of COVID-19 was not associated with abnormal spirometry results in children 6 months after infection [20]. However, the sample size of that study was small (*n* = 40). They also found that undernourished status was associated with abnormal spirometry results in children after COVID-19 (*p* = 0.028; OR 5.13; 95% CI, 1.19–22.11) [20]. This finding is consistent with our results, which identified undernutrition as the most significant factor influencing respiratory function after COVID-19. A cohort study in England involving the general population also found that undernutrition is a risk factor for restrictive-type lung disorders in children [18]. Undernutrition may impair lung function by increasing susceptibility to recurrent respiratory infections, which can lead to fibrosis and reduced breathing capacity [18]. Furthermore, a study in Arizona reported that toddlers with poor nutritional status were at risk of developing restrictive-type lung disorders. Nutritional deficiencies during growth and development may disturb lung parenchyma formation and maturation [19].

In our study, a history of hospitalization was not significantly associated with decreased respiratory function. This contrasts with findings by from Ipek et al., who observed that reduced lung function was significantly correlated with both the history and duration of hospitalization [18,21]. This discrepancy may be due to differences in hospitalization and discharge criteria among institutions treating children with COVID-19. Our hospital, being a national referral center, likely applied stricter hospitalization criteria. Moreover, discharge did not require a negative PCR result—patients with stable clinical conditions were allowed to continue self-quarantine at home. Because most of patients had comorbidities, many presented in critical condition and did not survive, and therefore were not included in the analysis. The small number of subjects with a history of critical COVID-19 may also explain why a history of ventilator support was not significantly associated with impaired lung function, even though three of the four subject (75%) who had received ventilator support showed abnormal spirometry results. Ventilator use can cause ventilator-induced lung injury (VILI) [30], which is associated with extensive inflammation, severe lung damage, and extensive fibrosis, potentially leading to long-term respiratory disability [30]. Studies in adults have shown that a median duration of 35 days of ventilator use is associated with lung fibrosis [31]; however, similar studies in children are lacking.

Although we attempted to minimize confounding by excluding children with pre-existing chronic lung disease or persistent asthma, we recognize that other factors during the 12–36 months interval (such as respiratory infections or allergies) could also have influenced pulmonary function. This limitation highlights the need for prospective longitudinal studies with baseline and follow-up spirometry to more definitively establish causality.

## 5. Conclusions

Our study is the first in Indonesia to evaluate respiratory function in children after COVID-19 using spirometry. The variables included in analysis encompassed not only COVID-19 severity but also other factors, such as comorbidities, nutritional status, length of hospital stay, history of ventilator support, and the presence of persistent symptoms. Moreover, our sample size of 100 participants is relatively large compared with similar studies.

Despite these strengths, several limitations should be noted. First, as a retrospective study, it is susceptible to information bias. Second, we did not perform additional diagnostic tests to evaluate lung pathology or assess perfusion components. Third, the absence of baseline spirometry data limited our ability to fully exclude the influence of other conditions that may have developed between the initial COVID-19 infection and spirometry assessment. Future prospective studies with serial lung function testing are needed to distinguish the effects of SARS-CoV-2 infection from other potential contributing factors.

In conclusion, nearly half of the children with a history of COVID-19 demonstrated impaired pulmonary function, predominantly a restrictive pattern. Persistent symptoms, undernourished status, and moderate-to-critical COVID-19 severity were significantly associated with abnormal spirometry results. These findings emphasize the importance of long-term respiratory monitoring in children after COVID-19, particularly among vulnerable groups. Further prospective studies with baseline and follow-up assessments are needed to confirm these associations and clarify the underlying mechanisms.

## Figures and Tables

**Figure 1 medicina-61-02136-f001:**
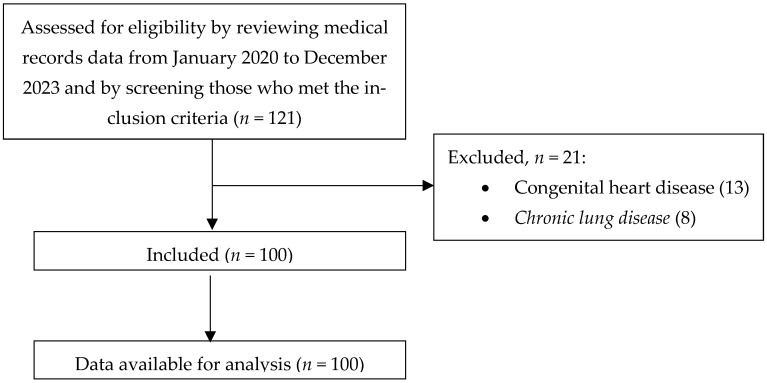
Flow chart of participants. *Created by the authors*.

**Figure 2 medicina-61-02136-f002:**
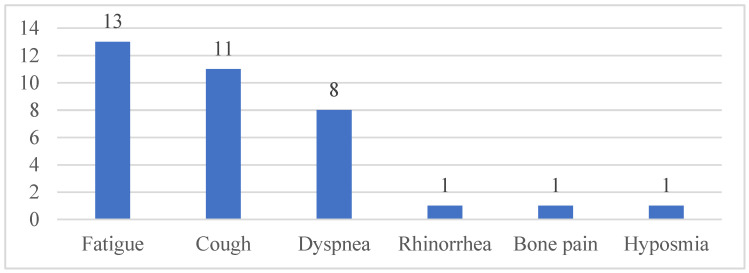
Frequency of each persistent symptom (*n*; %).

**Table 1 medicina-61-02136-t001:** Demographic and clinical characteristics of subjects.

Subjects Characteristics (*n* = 100)	N	%
**Gender**		
Male	47	47
Female	53	53
**Age (years)**		
5–<12	48	48
≥12–18	52	52
**Type of comorbidity**		
Obesity	27	27
Undernourishment	19	19
Chronic kidney disease	16	16
Hematology disorder	6	6
Allergy	4	4
Immunology disorder	3	3
Endocrine Disorder	2	2
Tuberculosis	2	2
Asthma	3	3
Neurological Disorder	2	2
Others	7	7
**Number of comorbidities**		
None	23	23
1 comorbidity	56	56
≥2 comorbidities	20	20
**Clinical classification of COVID-19**		
Asymptomatic	4	4
Mild	73	73
Moderate	16	16
Severe	3	3
Critical	4	4
**History of ventilator support**		
No	96	96
Yes	4	4
**Length of stay**		
Not hospitalized	78	78
<48 h	3	3
≥48 h	19	19
**Time interval from recovery to spirometry**		
6–12 months	6	6
>12–36 months	65	65
>36 months	29	29

**Table 2 medicina-61-02136-t002:** Bivariate analysis between independent variables and spirometry results.

Variable	Spirometry Result, *n* (%)			RR (95% CI)
	Abnormal	Normal	*p*	
**Clinical classification of COVID-19**				
Moderate–severe–critical	17 (73.9)	6 (26.1)	**0.002**	1.96 (1.35–2.86)
Asymptomatic–mild	29 (37.7)	48 (62.3)		
**Gender**				
Male	20 (42.6)	27 (57.4)	0.515	0.86 (0.56–1.33)
Female	26 (49.1)	27 (50.9)		
**Age (years old)**				
≥12–18	26 (50)	26 (50)	0.404	1.20 (0.78–1.85)
5–<12	20 (41.7)	28 (58.3)		
**Obesity**				
Yes	9 (33.3)	18 (66.7)	0.122	0.66 (0.37–1.17)
No	37 (50.7)	36 (49.3)		
**Undernourished**				
Yes	14 (73.7)	5 (26.3)	**0.007**	1.86 (1.27–2.73)
No	32 (39.5)	49 (60.5)		
**Chronic kidney disease**				
Yes	10 (62.5)	6 (37.5)	0.148	1.46 (0.93–2.23)
No	36 (42.9)	48 (57.1)		
**Hematology disorder**				
Yes	4 (66.7)	2 (33.7)	0.410 *	1.49 (0.81–2.74)
No	42 (44.7)	52 (55.3)		
**Allergy**				
Yes	1 (25.0)	3 (75)	0.622 *	0.53 (0.09–2.95)
No	45 (46.9)	51 (53.1)		
**Immunology disorder**				
Yes	1 (33.3)	2 (66.7)	1.00 *	0.72 (0.14–3.61)
No	45 (46.4)	52 (53.6)		
**Endocrine disorder**				
Yes	1 (50.0)	1 (50.0)	1.00 *	1.09 (2.68–4.42)
No	45 (45.9)	53 (54.1)		
**Tuberculosis**				
Yes	2 (100)	0 (0)	0.209 *	2.27 (1.79–2.77)
No	44 (44.9)	54 (55.1)		
**Asthma**				
Yes	1 (33.3)	2 (66.7)	1.00 *	0.72 (0.14–3.61)
No	45 (46.4)	52 (53.6)		
**Neurological disorder**				
Yes	2 (100)	0 (0)	0.209 *	2.23 (1.79–2.77)
No	44 (44.9)	54 (55.1)		
**Others**				
Yes	3 (42.9)	4 (57.1)	1.00 *	0.93 (0.38–2.24)
No	43 (46.2)	50 (53.8)		
**Comorbid**				
Yes	35 (45.5)	42 (54.5)	0.841	0.95 (0.58–1.55)
No	11 (47.8)	12 (52.2)		
**History of ventilator support**				
Yes	3 (75)	1 (25)	0.331 *	1.67 (0.91–3.07)
No	43 (44.8)	53 (55.2)		
**History of hospitalization**				
Yes	14 (63.6)	8 (36.4)	0.060	1.55 (1.02–2.34)
No	32 (41)	46 (59)		
**Time interval from recovery to spirometry (months)**				
≤12	3 (50.0)	3 (50.0)	1.00 *	1.1 (0.47–2.56)
>12	43 (43.2)	51 (50.8)		

* Fischer test.

**Table 3 medicina-61-02136-t003:** Multivariate logistic regression analysis of factors associated with impaired spirometry results.

Variable	*p*	aOR * (CI ** 95%)
Undernourished state	**0.002**	5.64 (1.89–16.85)
Clinical classification of COVD-19	**0.006**	5.18 (1.59–16.89)
Obesity	0.58	0.73 (0.24–2.22)
Chronic kidney disease	0.31	1.87 (0.55–6.38)
History of hospitalization	0.53	1.45 (0.45–4.74)

* aOR: Adjusted odds ratio. **** CI: Confidence interval**.

## Data Availability

The original contributions presented in this study are included in the article/Appendix A. Further inquiries can be directed to the corresponding author.

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
