# Peer review of "Pulmonary Function and Associated Prognostic Factors in Children After COVID-19: A Retrospective Cohort Study"

_medicina, 2025, doi:10.3390/medicina61122136_

Round 1

Reviewer 1 Report

Comments and Suggestions for Authors

I have reviewed the manuscript entitled "Prognostic Factors for Long Term Respiratory Function in Children After COVID-19 " written by Mega Septiana, Nastiti Kaswandani, Hindra Irawan Satari, Irene Yuniar, Adhi Teguh Perma Iskandar, Henny Adriani Puspitasari . The manuscript describes the main sequelae after COVID, persistent on long term evaluation in cohort of 100 children. The results of the study refer to pulmonary function, persistent symptoms and prognostic factors for long term respiratory function. The main limitation of the study regards the lack of control group/lack of prior-to-COVID investigation. If the hypothesis of the study is that COVID leads to long term pulmonary function impairment, then the results do not support the conclusions. In order to conclude that impaired spirometry is caused by COVID there must be a more detailed study protocol. In 12-36 months (the time between COVID and spirometry) many other factors can influence the pulmonary function (including allergies, other pulmonary infections, etc).  How were these factors excluded? Also, a comparison between 2 spirometry evaluations (before and after COVID) could better conclude the differences). 

Introduction is relatively short, data can be added regarding the discussed topics (undernourishment, pulmonary evaluation, COVID sequelae, Spirometry). The authors mention the aim of the study and also the hypothesis: "The aim of this study was to describe the lung function in children after COVID-19 and to determine the prevalence, characteristics, and also the prognostic factors that influence long term respiratory function in children after COVID-19. We hypothesize that children may experience impaired pulmonary function following COVID-19 infection. "

Material and methods section:

What were the inclusion criteria and exclusion criteria (how was "decreased lung function prior to COVID" evaluated?, How what the diagnosis established "had underlying disease
that affect long term lung function (e.g severe persistent asthma)"). Please describe in detail how you selected the study group. I would recommend for the Material and Methods section, some subtitles: Study design, Participants, Respiratory evaluation (clinical and spirometry - what technique, devise, how many evaluations, interpretation), Statistical analysis, Ethical approval.  how was undernourishment/obesity established? How was decided when to come for spirometry? Did the researchers ask participants to come for evaluation? Detail the clinical classification of initial disease (COVID).

Results

What is the prevalence of prognostic factors

Supplementary Table S1 contains valuable information that could be described within the text.

What does clinical classification in Table 3 refer to? What is the significance of the p value for clinical classification?

Discussions:

Lines 174-186, 195-201 describe theoretical details bout COVID and may belong to Introduction section.

"Line 228 - 229: "From multivariate analysis, we found that moderate-severe-critical COVID-19 and undernourished state were significantly influenced spirometry result in children after COVID-19"  refers to Table 3? Please explain Table 3 in the results section, change the name of the Table so the reader can understand its content and add legend with abbreviations. Also correct the above sentence "were influenced by or influenced".

What are the conclusions of the study? Is the hypothesis confirmed by the results?

Overall the manuscript describes the findings in clinical and respiratory evaluation (spirometry) in children after COVID. Is doesn`t have a strict study protocol to establish prognostic factors or details that COVID in the only variable that could interact with respiratory function. I would recommend changing the title in order to reflect the main findings of the study.

How was the study designed? the ethical approval is dated in 2023, the  retrospective study included patients between January-May 2024 (line 51), medical records from January 2020 - December 2023 (Figure 1)?

Please revise the spelling in entire document: line 69 "can be explained", Figure 2, line 216: "dyspneu", line 119 "There was / were" 

Author Response

For research article

Response to Reviewer X Comments

1. Summary

2. Questions for General Evaluation

Reviewer’s Evaluation

Response and Revisions

Does the introduction provide sufficient background and include all relevant references?

Can be improved

Thank you for the evaluation. I will add more background to the introduction in the manuscript revision.

Is the research design appropriate?

Can be improved

Thank you for the evaluation. I already add more detail in reseaech design according to reviewer 1.

Are the methods adequately described?

Must be improved

Thank you for the evaluation. I already add more detail in methods section in the manuscript revision.

Are the results clearly presented?

Can be improved

Thank you for the evaluation. I already made adjustment for presenting result in the manuscript revision.

Are the conclusions supported by the results?

Must be improved

Thank you for the evaluation. I will make adjustment in conclusion in the manuscript revision.

3. Point-by-point response to Comments and Suggestions for Authors

Comments 1:

If the hypothesis of the study is that COVID leads to long term pulmonary function impairment, then the results do not support the conclusions. In order to conclude that impaired spirometry is caused by COVID there must be a more detailed study protocol.

Response 1:

We thank the reviewer for this important comment. We agree that our retrospective design without baseline spirometry before COVID-19 does not allow us to establish causality. Therefore, we have revised the wording in both the Abstract and Discussion to emphasize association rather than causation.

Revisions in the manuscript:

  • Page 1, lines 26–27 (Abstract conclusion):
    Revised from:
    “Persistent symptoms, undernourished status and moderate-severe-critical of COVID-19 are the prognostic factors for poor long term respiratory function in post-COVID-19 pediatric patients.”
    to:
    “Persistent symptoms, undernourished status, and moderate-to-critical severity of COVID-19 were found to be associated with impaired long-term respiratory function in post-COVID-19 pediatric patients. Further prospective studies are needed to confirm these findings and clarify causal mechanisms”
  • Page 8, lines 264 (Discussion, Conclusion paragraph):
    Revised from:
    “… were significantly influenced spirometry result in children after COVID-19 …”
    to:
    “…were significantly associated with impaired spirometry results in children after COVID-19. Our results underscore the importance of early identification of at-risk children, and future longitudinal studies with baseline and follow-up pulmonary function assessments are encouraged to confirm these associations and explore underlying mechanisms.”

Comments 2:

In 12–36 months (the time between COVID and spirometry) many other factors can influence pulmonary function (including allergies, other pulmonary infections, etc). How were these factors excluded? Also, a comparison between 2 spirometry evaluations (before and after COVID) could better conclude the differences).

Response 2: We thank the reviewer for this insightful comment. We agree that multiple factors could potentially influence pulmonary function in the 12–36 months after recovery. To minimize confounding, we excluded patients with known chronic lung disease (e.g., bronchopulmonary dysplasia, congenital lung anomalies, or persistent asthma) and those with underlying conditions that could affect long-term lung function. Information regarding comorbidities, hospitalization, and clinical course was verified through both medical records and parental reports to reduce recall bias.

We acknowledge that the absence of baseline spirometry prior to COVID-19 is a limitation of our study, and thus causal inference cannot be established. However, our findings still provide valuable insight into the prevalence and associated factors of impaired lung function in post-COVID-19 children. We have revised the Discussion and Limitations sections to clarify this point and emphasize that prospective longitudinal studies with serial spirometry (before and after infection) are needed to better establish causality and differentiate the impact of COVID-19 from other potential contributing factors.

Revisions in the manuscript:

  • Page 9, Discussion, (Prognostic factors): Added:
    “Although we attempted to minimize confounding by excluding children with pre-existing chronic lung disease or persistent asthma, we recognize that other factors occurring during the 12–36 month interval (such as respiratory infections or allergies) could also influence pulmonary function. This limitation highlights the need for prospective longitudinal studies with baseline and follow-up spirometry to more definitively establish causality.”
  • Page 9, Strength and Limitation section: Expanded:
    “Our retrospective design without baseline spirometry did not allow us to fully exclude the influence of other conditions that may have occurred between COVID-19 infection and spirometry assessment. Future prospective studies with serial lung function testing are needed to disentangle the effects of SARS-CoV-2 from other potential contributing factors.”

Comments 3:

Introduction is relatively short, data can be added regarding the discussed topics (undernourishment, pulmonary evaluation, COVID sequelae, Spirometry).

Response 3:

We thank the reviewer for this valuable suggestion. We have revised and expanded the Introduction section to include additional background on the impact of undernutrition on lung development, the role of spirometry in evaluating pulmonary function, and recent evidence regarding COVID-19 sequelae in children. These revisions strengthen the rationale for our study.

Comments 4:

The authors mention the aim of the study and also the hypothesis: "The aim of this study was to describe the lung function in children after COVID-19 and to determine the prevalence, characteristics, and also the prognostic factors that influence long term respiratory function in children after COVID-19. We hypothesize that children may experience impaired pulmonary function following COVID-19 infection.

Response 4:

We thank the reviewer for highlighting this point. Our study was designed to describe lung function outcomes in children after COVID-19 and to analyze potential prognostic factors. We agree that the wording of our hypothesis could be interpreted as implying causality. To avoid misinterpretation and to remain consistent with our retrospective study design, we have revised the hypothesis in the Introduction to emphasize the exploration of associations between COVID-19 history and long-term pulmonary function.

Comments 5:
What were the inclusion criteria and exclusion criteria (how was "decreased lung function prior to COVID" evaluated?, How was the diagnosis established "had underlying disease that affect long term lung function (e.g. severe persistent asthma)"). Please describe in detail how you selected the study group.

Response 5:
We thank the reviewer for pointing this out. We have now expanded the Methods section to provide a more detailed description of the inclusion and exclusion criteria and the process of subject selection. Specifically:

  • Evaluation of decreased lung function prior to COVID-19: We reviewed the patients’ medical records and parental reports. Children with documented history of chronic respiratory symptoms (e.g., persistent dyspnea, exercise limitation) or prior spirometry indicating abnormal lung function before SARS-CoV-2 infection were excluded.
  • Underlying diseases affecting long-term lung function: These were defined based on established clinical criteria. For example, severe persistent asthma was diagnosed according to GINA guidelines, while congenital heart disease or chronic lung disease (such as bronchopulmonary dysplasia, lung hypoplasia, or lung cysts) were confirmed from specialist medical records.
  • Study group selection: All patients aged 5–18 years with a confirmed history of COVID-19 between January 2020 and December 2023 were screened using hospital medical records. Eligible patients who met the inclusion criteria and had no exclusion conditions were invited, and informed consent was obtained from parents/guardians.

Revisions in the manuscript:

  • Page 2, Methods, paragraph 1: Expanded to read:
    “Patients with decreased lung function prior to COVID-19 were excluded based on review of medical records and parental reports of persistent respiratory symptoms or abnormal pre-COVID spirometry results, when available. Underlying diseases affecting long-term lung function (e.g., severe persistent asthma diagnosed according to GINA criteria, congenital heart disease, or chronic lung disease such as bronchopulmonary dysplasia or lung hypoplasia) were confirmed from specialist medical records. Eligible patients were identified from hospital records of children aged 5–18 years with confirmed COVID-19 between January 2020 and December 2023 and were included after written informed consent from parents/guardians.”

Comments 6:
I would recommend for the Material and Methods section, some subtitles: Study design, Participants, Respiratory evaluation (clinical and spirometry - what technique, devise, how many evaluations, interpretation), Statistical analysis, Ethical approval. How was undernourishment/obesity established? How was decided when to come for spirometry? Did the researchers ask participants to come for evaluation? Detail the clinical classification of initial disease (COVID).

Response 6:
We thank the reviewer for these helpful suggestions. We have revised the Materials and Methods section to include subtitles for clarity and added more detail regarding participant selection, nutritional assessment, spirometry procedures, and disease classification. Specifically:

  • Study design: now clearly stated as an observational analytical study with retrospective cohort design.
  • Participants: inclusion and exclusion criteria clarified, including how decreased lung function prior to COVID-19 and underlying diseases were assessed (based on medical records, parental reports, and clinical diagnosis as per guidelines).
  • Nutritional status: undernutrition and obesity were determined using WHO growth standards, with BMI-for-age z-score < -2SD classified as undernourished and > +2SD classified as obese.
  • Respiratory evaluation: spirometry was performed at least 6 months post-COVID recovery, using standardized equipment (e.g., Jaeger MasterScreen™) according to ATS/ERS guidelines. Each child performed at least three acceptable maneuvers, and results were interpreted based on predicted values for age, sex, and height. Clinical classification of COVID-19 was determined from medical records according to Indonesian Pediatric Society/WHO criteria (asymptomatic, mild, moderate, severe, critical).
  • Scheduling of spirometry: participants were invited to come for evaluation through outpatient follow-up clinics, ensuring that all had recovered for at least 6 months prior.
  • Statistical analysis and Ethical approval: moved into separate subheadings for clarity.

Revisions in the manuscript:

  • Methods section: reorganized with subtitles (Study design, Participants, Respiratory evaluation, Statistical analysis, Ethical approval).
  • Participants subsection: Expanded to read:
    “Patients with decreased lung function prior to COVID-19 were excluded based on review of medical records and parental reports of persistent respiratory symptoms or abnormal pre-COVID spirometry results, when available. Underlying diseases affecting long-term lung function (e.g., severe persistent asthma diagnosed according to GINA criteria, congenital heart disease, or chronic lung disease such as bronchopulmonary dysplasia or lung hypoplasia) were confirmed from specialist medical records. Nutritional status was determined using WHO BMI-for-age z-scores, with < -2SD defined as undernourished and > +2SD defined as obese.”
  • Respiratory evaluation subsection: Added:
    “Pulmonary function was assessed using spirometry (HI-901, Chest MI, Inc., Japan) following the American Thoracic Society/European Respiratory Society (ATS/ERS) 2019 recommendations. Each subject performed at least three technically acceptable maneuvers, and the highest values were recorded. Predicted values were adjusted for age, sex, and height. Lung function was categorized as normal, obstructive, restrictive, or mixed pattern based on FEV1 and FVC predicted values adjusted for age, sex, and height. Clinical classification of COVID-19 (asymptomatic, mild, moderate, severe, critical) was established from medical records according to WHO/Indonesian Pediatric Society guidelines. Participants were scheduled for spirometry at least 6 months after confirmed recovery and were invited to attend through outpatient follow-up visits.”

Comment 7:

What is the prevalence of prognostic factors?

Response 7:
We thank the reviewer for this important point. The prevalence of the significant prognostic factors was added to the Results section for clarity. Specifically, undernourishment was found in 19% of the participants, and 23% of children experienced moderate-to-critical COVID-19. These variables were significantly associated with impaired lung function.

Revision in manuscript:

  • Results (Prognostic factors subsection):
    “Among the study population, 19% of children were undernourished and 23% had experienced moderate-to-critical COVID-19. Both factors were significantly associated with impaired spirometry results in multivariate analysis.”

Comments 8:

Supplementary Table S1 contains valuable information that could be described within the text.

Response 8:
We agree with the reviewer that the information from Supplementary Table S1 strengthens the Results section. We have therefore included the main findings of this supplementary material in the text, particularly regarding the mean and median spirometry values.

Revision in manuscript:

  • Results (Pulmonary function subsection):
    “Detailed spirometry parameters are presented in Supplementary Table S1. In brief, the mean predicted values of most parameters were above 80%, except for FEV1 which showed a median of 77.9%, indicating a mild reduction in expiratory flow in a proportion of subjects.”

Comments 9:

What does clinical classification in Table 3 refer to? What is the significance of the p value for clinical classification?

Response 9:
We appreciate the reviewer’s request for clarification. “Clinical classification” in Table 3 refers to the severity of the initial COVID-19 episode, categorized as asymptomatic/mild versus moderate-to-critical, based on WHO and Indonesian Pediatric Society guidelines. The significant p-value (p = 0.006) in the multivariate analysis indicates that children with a history of moderate-to-critical COVID-19 had a significantly higher risk of impaired pulmonary function compared to those with asymptomatic or mild disease (aOR 5.18; 95% CI 1.59–16.89).

Revision in manuscript:

  • Results (Prognostic factors subsection):
    “Clinical classification referred to the initial severity of COVID-19 (asymptomatic/mild vs. moderate-to-critical). A significant association was observed (p = 0.006), with children who experienced moderate-to-critical COVID-19 showing more than fivefold higher odds of impaired pulmonary function compared to those with asymptomatic or mild disease.”

Comments 10: Lines 174–186, 195–201 describe theoretical details about COVID and may belong to Introduction section.

Response 10:
We thank the reviewer for this observation. We have revised the manuscript by moving these theoretical details about COVID-19 pathophysiology from the Discussion section to the Introduction, where they provide more appropriate background context.

Revision in manuscript:

  • Moved lines 174–186, 195–201 (Discussion, Pulmonary function subsection) relocated into Introduction, paragraph 3.

Comments 11:

Line 228–229: "From multivariate analysis, we found that moderate-severe-critical COVID-19 and undernourished state were significantly influenced spirometry result in children after COVID-19" refers to Table 3? Please explain Table 3 in the results section, change the name of the Table so the reader can understand its content and add legend with abbreviations. Also correct the above sentence "were influenced by or influenced".

Response 11:
We appreciate the reviewer’s suggestion. We have clarified that this sentence refers to Table 3, explained the table content in the Results section, and changed the table title for clarity. We also added a legend with explanations of abbreviations. The wording of the sentence has been revised to “were significantly associated with impaired spirometry results.”

Revisions in manuscript:

  • Results, Prognostic factors subsection:
    “From multivariate analysis, we found that moderate-to-critical COVID-19 severity and undernourished status were significantly associated with impaired spirometry results in children after COVID-19 (Table 3).”
  • Table 3 title changed to: “Multivariate logistic regression analysis of factors associated with impaired spirometry results.”
  • Legend added to Table 3: aOR: adjusted odds ratio; CI: confidence interval.

Comments 12:

What are the conclusions of the study? Is the hypothesis confirmed by the results?

Response 12:
We thank the reviewer for this important question. The hypothesis that children may experience impaired pulmonary function following COVID-19 was supported, as 47% of participants showed abnormal spirometry, mainly restrictive pattern. Furthermore, our study identified undernutrition and moderate-to-critical COVID-19 severity as significant factors associated with these impairments. We have revised the Conclusion section in both the Abstract and Discussion to make this clearer.

Revisions in manuscript:

  • Abstract conclusion:
    “Persistent symptoms, undernourished status, and moderate-to-critical severity of COVID-19 were found to be associated with impaired long-term respiratory function in post-COVID-19 pediatric patients. Further prospective studies are needed to confirm these findings and clarify causal mechanisms.”
  • Discussion conclusion: updated accordingly.

Comments 13:

Overall the manuscript describes the findings in clinical and respiratory evaluation (spirometry) in children after COVID. It doesn’t have a strict study protocol to establish prognostic factors or details that COVID is the only variable that could interact with respiratory function. I would recommend changing the title in order to reflect the main findings of the study.

Response 13:
We agree with the reviewer that causality cannot be inferred due to the retrospective design and lack of pre-COVID spirometry. To better reflect the scope and findings, we have revised the title to:

New Title:
“Pulmonary Function and Associated Prognostic Factors in Children after COVID-19: A Retrospective Cohort Study”

This emphasizes the description of lung function and associated factors, without implying causality.

Comments 14:

How was the study designed? The ethical approval is dated in 2023, the retrospective study included patients between January–May 2024 (line 51), medical records from January 2020 – December 2023 (Figure 1)?

Response 14:
We thank the reviewer for pointing out this confusion. To clarify:

  • The study design was a retrospective cohort.
  • Patient recruitment and spirometry evaluation were conducted between January–May 2024.
  • Clinical data were obtained retrospectively from medical records covering January 2020–December 2023.
  • Ethical approval was obtained in late 2023, prior to patient recruitment in 2024.

We have revised the Methods section to make this timeline explicit.

Revisions in manuscript:

  • Study Design subsection:
    “Eligible patients with confirmed COVID-19 between January 2020 and December 2023 were identified from hospital records, and follow-up spirometry evaluation was performed in 2024 after ethical approval (issued in late 2023).”

Comments 15:

Please revise the spelling in entire document: line 69 "can be explained", Figure 2, line 216: "dyspneu", line 119 "There was / were".

Response 15:
We thank the reviewer for this careful reading. We have corrected these spelling and grammar errors throughout the manuscript:

  • Line 69 was omitted due adjustment to reviewer
  • Figure 2, Line 216: corrected “dyspneu” to “dyspnea”.
  • Line 119: revised to “There were …”.
  • A full spell-check was also conducted to ensure consistency.

4. Response to Comments on the Quality of English Language

Point 1: The English is fine and does not require any improvement.

Response 1:

We thank the reviewer for this positive feedback. No changes were required regarding the language.

5. Additional clarifications

The data presented in this study are available on request from the corresponding author. The authors declare no conflicts of interest. No external funding was received for this research.

Reviewer 2 Report

Comments and Suggestions for Authors

The authors conducted a retrospective study to evaluate prognostic factors for long-term respiratory function in children after COVID-19. The study is well designed, and the results are thoughtfully discussed. There are a few questions and suggestions for the authors’ consideration:

  1. Methods Section
    a. Please provide the full spelling of all respiratory parameters (e.g., FVC and FEV₁). Please also include FEF₂₅ and FEF₅₀, since these are also mentioned in the results section.
    b. In the statistical methods section (lines 81-86), please include RR as well as a full spelling (Relative Risk), as it is an important measurement in the results.
  2. Results Section
    a. In Table 1, the categories and variables under each category are not clearly visualized. It may help to follow the format of Table 2 and bold each category. Also, please correct the spelling of “Clinal Classiciationi of COVID-19”.
    b. In Table S1, parameters are summarized using different formats. Some are median with range, and some are median with SD. Please clarify whether there is a rationale for this. If not, please be consistent. Also, some digits appear with commas instead of decimal points and please correct this.
    c. In Table 2, please clarify what the * symbol indicates for certain p-values.
    d. In Table 3, please consistently use the full spelling “Clinical classification of COVID-19” to match the terminology in earlier tables. For line 133 (** Fischer test), please clarify whether this corresponds to any “**” mark within the table.
  3. Discussion Section

a. In lines 160–162, the phrase “But fortunately…” is confusing. Given that diffusion parameters were not measured, should this instead read “unfortunately”? Also, please provide the full spelling of DLCO.

b. In lines 167–172, the authors summarized some literature with different observations. Is there any possible reason why these different results may happen?
c. In lines 212–213, please correct the phrase “decreased of FVC” to “decreased FVC”.
d. In line 230, “aOR 5,4” uses a comma instead of a decimal point. Please ensure consistency throughout the manuscript for this formatting issue.
e. In line 270, does “Strength and Limitation” fall within the discussion section? Please add a section number correspondingly.

Author Response

For research article

Response to Reviewer X Comments

1. Summary

2. Questions for General Evaluation

Reviewer’s Evaluation

Response and Revisions

Does the introduction provide sufficient background and include all relevant references?

Yes

Thank you for the evaluation. I already add more background to the introduction in the manuscript revision according to reviewer 1.

Is the research design appropriate?

Yes

Thank you for the evaluation. I already add more detail in reseaech design according to reviewer 1.

Are the methods adequately described?

Can be improved

Thank you for the evaluation. I already add more detail in methods section in the manuscript revision.

Are the results clearly presented?

Can be improved

Thank you for the evaluation. I already made adjustment for presenting result in the manuscript revision.

Are the conclusions supported by the results?

Yes

Thank you for the evaluation.

3. Point-by-point response to Comments and Suggestions for Authors

Comments 1:

Please provide the full spelling of all respiratory parameters (e.g., FVC and FEV₁). Please also include FEF₂₅ and FEF₅₀, since these are also mentioned in the results section.

Response 1:

We thank the reviewer for this comment. We have revised the Methods section to include the full spelling of all pulmonary function parameters: forced vital capacity (FVC), forced expiratory volume in one second (FEV₁), forced expiratory flow at 25% of FVC (FEF₂₅), and forced expiratory flow at 50% of FVC (FEF₅₀).

Revisions in the manuscript:

  • Respiratory evaluation subsection updated accordingly.

Comments 2:

In the statistical methods section (lines 81–86), please include RR as well as a full spelling (Relative Risk).

Response 2:
We agree with the reviewer. We have revised the Statistical Analysis subsection to include “RR (Relative Risk)” in full spelling.

Revision in manuscript:

  • Page 4, Statistical analysis subsection updated accordingly

Comments 3:

In Table 1, the categories and variables under each category are not clearly visualized. It may help to follow the format of Table 2 and bold each category. Also, please correct the spelling of “Clinal Classiciationi of COVID-19”

Response 3:

We thank the reviewer. Table 1 has been reformatted for clarity, with categories bolded and variables aligned consistently. The spelling error has also been corrected to “Clinical classification of COVID-19.”

Comments 4:

In Table S1, parameters are summarized using different formats. Some are median with range, and some are median with SD. Please clarify whether there is a rationale for this. If not, please be consistent. Also, some digits appear with commas instead of decimal points and please correct this.

Response 4:

We thank the reviewer for this observation. The use of different formats in Table S1 was intentional. Variables with normal distribution are presented as mean ± standard deviation (SD), while variables with non-normal distribution are presented as median with interquartile range (IQR). We have clarified this in the table legend. In addition, all formatting inconsistencies have been corrected, with decimal points used consistently instead of commas.

Revision in manuscript:

  • Table S1 legend updated to: “Data are presented as mean ± standard deviation (SD) for normally distributed variables, and median (interquartile range) for non-normally distributed variables.”
  • All commas replaced with decimal points for consistency.

Comments 5:
In Table 2, please clarify what the * symbol indicates for certain p-values.

Response 5:
We thank the reviewer for pointing this out. We have clarified the legend of Table 2 now states that “* indicates p-value obtained using Fisher’s exact test.”

Comments 6:
I would recommend for the Material and Methods section, some subtitles: Study design, Participants, Respiratory evaluation (clinical and spirometry - what technique, devise, how many evaluations, interpretation), Statistical analysis, Ethical approval. How was undernourishment/obesity established? How was decided when to come for spirometry? Did the researchers ask participants to come for evaluation? Detail the clinical classification of initial disease (COVID).

Response 6:
We thank the reviewer for pointing this out. We have corrected in Table 3 consistently uses “Clinical classification of COVID-19,” and the legend was revised to clarify that “**” indicates confidence interval, but ‘*’ in Table 2 refer to fischer’s exact test meanwhile in Table 3 refer to adjusted odds ratio.

Comment 7:

In lines 160–162, the phrase “But fortunately…” is confusing. Given that diffusion parameters were not measured, should this instead read “unfortunately”? Also, please provide the full spelling of DLCO.

Response 7:
We thank the reviewer for this important point. The phrase has been changed to “Unfortunately” and DLCO has been spelled out as “diffusing capacity of the lungs for carbon monoxide (DLCO)

Comments 8:

In lines 167–172, the authors summarized some literature with different observations. Is there any possible reason why these different results may happen?

Response 8:
We agree with the reviewer so we added possible explanations for the discrepancies, including differences in study design, timing of pulmonary evaluation after infection, severity of initial disease, and whether comorbidities were excluded.

  • Revision in manuscript: Discussion.

Comments 9:

In lines 212–213, please correct the phrase “decreased of FVC” to “decreased FVC”.

Response 9:
Corrected as suggested.

Comments 10:

In line 230, “aOR 5,4” uses a comma instead of a decimal point. Please ensure consistency throughout the manuscript for this formatting issue.

Response 10:
Corrected. All numerical data are now presented with decimal points.

Comments 11:

In line 270, does “Strength and Limitation” fall within the discussion section? Please add a section number correspondingly.

Response 11:
We thank the reviewer for this suggestion. To improve clarity, we have revised the structure of the manuscript so that Strengths and Limitations is now presented as a separate section following the Discussion, rather than as a subsection.

Revision in manuscript:

  • Section heading changed to “5. Strengths and Limitations”, placed after the Discussion.

4. Response to Comments on the Quality of English Language

Point 1: The English is fine and does not require any improvement.

Response 1:

We thank the reviewer for this positive feedback. No changes were required regarding the language.

5. Additional clarifications

The data presented in this study are available on request from the corresponding author. The authors declare no conflicts of interest. No external funding was received for this research.

Round 2

Reviewer 1 Report

Comments and Suggestions for Authors

The present manuscript has still no conclusions. Request 12 from initial review is not present in the revised manuscript conclusion section, only in the abstract part.

The document has improved after corrections.

In the discussion section, can it be possible to compare the incidence of decreased lung function after COVID (abnormal spirometry) with literature incidence of decreased lung function in normal children? Because this can perhaps emphasize the idea that COVID can lead to abnormal lung function.

Please state the hypothesis of the study clear in the end of Introduction.

Author Response

Thank you very much for your review and input. 

Reviewer Comment 1: The present manuscript has still no conclusions. Request 12 from initial review is not present in the revised manuscript conclusion section, only in the abstract part.

Response:
We thank the reviewer for this important note. We have now revised the manuscript to include a clear Conclusion section at the end of the paper, consistent with the abstract. The section now explicitly summarizes the main findings, emphasizes the clinical relevance, and highlights the need for further research.

Revised text (added at the end of manuscript):

  1. Conclusions
    In conclusion, we found that nearly half of children with a history of COVID-19 demonstrated impaired pulmonary function, predominantly a restrictive pattern. Persistent symptoms, undernourished status, and moderate-to-critical COVID-19 severity were significantly associated with abnormal spirometry results. These findings emphasize the importance of long-term respiratory monitoring in children after COVID-19, particularly among vulnerable groups. Further prospective studies with baseline and follow-up assessments are needed to confirm these associations and clarify the underlying mechanisms.

Reviewer Comment 2: In the discussion section, can it be possible to compare the incidence of decreased lung function after COVID (abnormal spirometry) with literature incidence of decreased lung function in normal children? Because this can perhaps emphasize the idea that COVID can lead to abnormal lung function.

Response:
We agree with the reviewer’s valuable suggestion. In the Discussion section, we have now added a comparison between the incidence of abnormal lung function in our post-COVID cohort (47%) with available literature on healthy children, where the prevalence of abnormal spirometry is reported to be much lower (generally <10% in community-based cohorts without chronic lung disease). This addition reinforces the interpretation that COVID-19 may contribute to the higher observed rate of impaired pulmonary function.

Revised text (added in Discussion, section 4.1 Pulmonary Function):

When compared with the general pediatric population without a history of COVID-19, the prevalence of abnormal spirometry in our cohort (47%) is considerably higher. Previous community-based studies in healthy school-aged children have reported abnormal lung function in less than 10% of cases, mostly related to undiagnosed asthma or transient respiratory infections rather than restrictive patterns.22-25 This contrast strengthens the notion that SARS-CoV-2 infection may contribute to the increased risk of restrictive lung impairment observed in our study population.

Reviewer Comment 3: Please state the hypothesis of the study clear in the end of Introduction.

Response:
Thank you for pointing this out. We have now revised the last paragraph of the Introduction to explicitly state the study hypothesis.

Revised text (added to the end of Introduction):

We hypothesize that children may experience impaired pulmonary function following COVID-19 infection, with undernutrition and a more severe disease course acting as major risk factors for long-term respiratory sequelae.

Reviewer 2 Report

Comments and Suggestions for Authors

The authors addressed all my questions and made corresponding changes. I do not have further questions at this point.

Author Response

Thank you very much for your review.
